# Alpha-1 antitrypsin inhibits TMPRSS2 protease activity and SARS-CoV-2 infection

Lukas Wettstein [1,16], Tatjana Weil [1,16], Carina Conzelmann[1,16], Janis A. Müller [1], Rüdiger Groß [1], Maximilian Hirschenberger[1], Alina Seidel[1], Susanne Klute[1], Fabian Zech[1], Caterina Prelli Bozzo[1], Nico Preising[2], Giorgio Fois [3], Robin Lochbaum[3], Philip Maximilian Knaff[4,5], Volker Mailänder[4,5], Ludger Ständker [2], Dietmar Rudolf Thal [6,7], Christian Schumann [8], Steffen Stenger[9], Alexander Kleger [10], Günter Lochnit [11], Benjamin Mayer[12], Yasser B. Ruiz-Blanco[13], Markus Hoffmann [14,15], Konstantin M. J. Sparrer [1], Stefan Pöhlmann [14], Elsa Sanchez-Garcia [13], Frank Kirchhoff [1], Manfred Frick [3] & Jan Münch [1,2✉]

SARS-CoV-2 is a respiratory pathogen and primarily infects the airway epithelium. As our knowledge about innate immune factors of the respiratory tract against SARS-CoV-2 is limited, we generated and screened a peptide/protein library derived from bronchoalveolar lavage for inhibitors of SARS-CoV-2 spike-driven entry. Analysis of antiviral fractions revealed the presence of $\alpha_1$-antitrypsin ($\alpha_1$AT), a highly abundant circulating serine protease inhibitor. Here, we report that $\alpha_1$AT inhibits SARS-CoV-2 entry at physiological concentrations and suppresses viral replication in cell lines and primary cells including human airway epithelial cultures. We further demonstrate that $\alpha_1$AT binds and inactivates the serine protease TMPRSS2, which enzymatically primes the SARS-CoV-2 spike protein for membrane fusion. Thus, the acute phase protein $\alpha_1$AT is an inhibitor of TMPRSS2 and SARS-CoV-2 entry, and may play an important role in the innate immune defense against the novel coronavirus. Our findings suggest that repurposing of $\alpha_1$AT-containing drugs has prospects for the therapy of COVID-19.

[1] Institute of Molecular Virology, Ulm University Medical Center, Ulm, Germany. [2] Core Facility Functional Peptidomics, Ulm University Medical Center, Ulm, Germany. [3] Institute of General Physiology, Ulm University, Ulm, Germany. [4] Dermatology Clinic, University Medicine Mainz, Mainz, Germany. [5] Max-Planck-Institute for Polymer Research, Mainz, Germany. [6] Laboratory of Neuropathology, Department of Imaging and Pathology, KU-Leuven and Department of Pathology, UZ-Leuven, Leuven, Belgium. [7] Laboratory of Neuropathology, Institute of Pathology, Ulm University, Ulm, Germany. [8] Pneumology, Thoracic Oncology, Sleep and Respiratory Critical Care Medicine, Clinics Allgäu, Kempten and Immenstadt, Germany. [9] Institute for Microbiology and Hygiene, Ulm University Medical Center, Ulm, Germany. [10] Department of Internal Medicine 1, Ulm University Hospital, Ulm, Germany. [11] Institute of Biochemistry, Justus-Liebig University Giessen, Giessen, Germany. [12] Institute for Epidemiology and Medical Biometry, Ulm University, Ulm, Germany. [13] Computational Biochemistry, Center of Medical Biotechnology, University of Duisburg-Essen, Essen, Germany. [14] Infection Biology Unit, German Primate Center- Leibniz institute for Primate Research, Göttingen, Germany. [15] Faculty of Biology and Psychology, Georg-August-University, Göttingen, Germany. [16] These authors contributed equally: Lukas Wettstein, Tatjana Weil, Carina Conzelmann. ✉email: jan.muench@uni-ulm.de

SARS-CoV-2 is mainly transmitted through inhalation of droplets and aerosols and subsequent infection of cells of the respiratory tract[1]. In many cases, infection is limited to the upper airways resulting in no or mild symptoms. Severe disease is caused by viral dissemination to the lungs ultimately resulting in acute respiratory distress syndrome, cytokine storm, multi-organ failure, septic shock, and death[2,3]. The airway epithelium acts as a frontline defense against respiratory pathogens via the mucociliary clearance and its immunological functions[4]. The epithelial lining fluid is rich in innate immunity peptides as well as proteins with antibacterial and antiviral activity, such as lysozyme, lactoferrin or defensins[5]. Currently, our knowledge about innate immune defense mechanisms against SARS-CoV-2 in the respiratory tract is limited.

To identify endogenous antiviral peptides and proteins, we previously generated peptide/protein libraries from body fluids and tissues and screened the resulting fractions for antiviral factors[6]. This approach allowed to identify novel modulators of HIV-1[7], CMV[8], and HSV-2[9] infection, with prospects for clinical development as antiviral drugs[10]. In this work, we set out to identify factors of the respiratory tract that block SARS-CoV-2 infection. Screening a peptide/protein library derived from bronchoalveolar lavage allowed to identify $\alpha_1$-antitrypsin ($\alpha_1$AT), a highly abundant circulating serine protease inhibitor, as SARS-CoV-2 entry inhibitor. We show that $\alpha_1$AT suppresses viral replication in cell lines and human airway epithelial cultures and that it binds and inactivates the serine protease TMPRSS2, which enzymatically primes the SARS-CoV-2 spike protein for membrane fusion. Hence, $\alpha_1$AT

blocks SARS-CoV-2 entry by inhibiting TMPRSS2, and may play an important role in the innate immune defense against the novel coronavirus. Our findings suggest that repurposing of $\alpha_1$AT-containing drugs has prospects for treatment of COVID-19.

## Results

**Identification of $\alpha_1$AT as SARS-CoV-2 inhibitor.** To discover endogenous antiviral peptides and proteins, we extracted polypeptides from 6.5 kg of homogenized human lung or 20 liters of pooled bronchoalveolar lavage (BAL), and separated them by chromatographic means. The corresponding fractions were added to human epithelial colorectal carcinoma (Caco2) cells and the cells were inoculated with luciferase expressing lentiviral pseudoparticles carrying the SARS-CoV-2 spike protein[11]. None of the fractions of the lung library suppressed infection (Supplementary Fig. 1a). In contrast, fractions 42–45 of the BAL library prevented SARS-CoV-2 spike driven entry with an efficiency comparable to that of 10 µM EK1, a coronavirus spike-specific peptide fusion inhibitor[12] (Fig. 1a). Titration of BAL fractions 42–45 onto Caco2 cells confirmed dose-dependent inhibition of SARS-CoV-2 spike pseudoparticles (Supplementary Fig. 1b).

To isolate the antiviral factor responsible for blocking spike-driven entry, the BAL mother fraction 42 was further separated chromatographically and the resulting sub-fractions analyzed for antiviral activity. As shown in Fig. 1b, sub-fractions 42_3 to 42_8 and 42_57 reduced, and sub-fraction 42_55 almost completely prevented host cell entry of SARS-CoV-2 spike pseudoparticles.

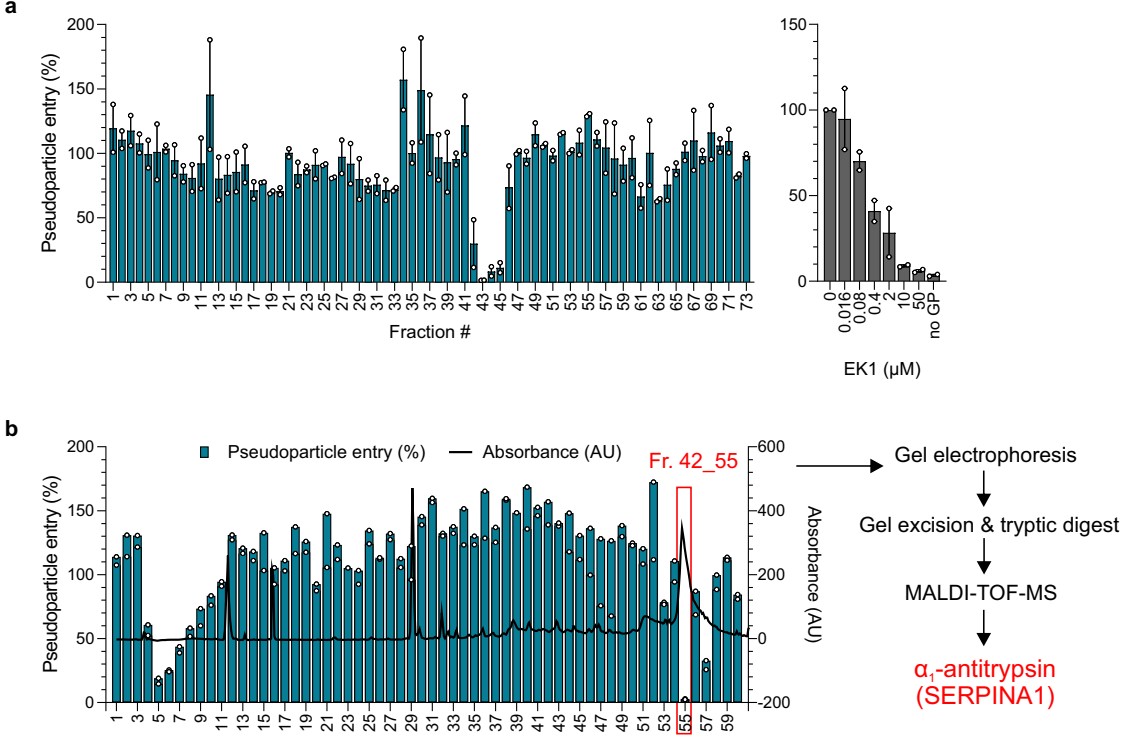

**Fig. 1 Identification of $\alpha_1$AT as SARS-CoV-2 inhibitor. a** Caco2 cells were treated with peptide/protein containing fractions of the bronchoalveolar lavage library (or EK1 peptide as inhibitor control) and transduced with luciferase-encoding lentiviral SARS-CoV-2 spike pseudoparticles. **b** Caco2 cells were treated with subfractions of mother fraction 42 (see **a**) of the bronchoalveolar lavage library and transduced with lentiviral SARS-CoV-2 spike pseudoparticles. Blue columns represent pseudoparticle entry and black line absorbance at 280 nm of the corresponding fraction. Transduction rates in **a** and **b** were determined 2 days after the addition of pseudoparticles by measuring luciferase activities in cell lysates. The means ± SEM from $n = 2$ (**a**) or individual datapoints from $n = 1$ (**b**) independent experiments are shown, each performed in biological duplicates. The active fraction 42_55 (red box) was analyzed via gel electrophoresis, gel excision, tryptic digest, and MALDI-TOF-MS to identify $\alpha_1$AT, a serine protease inhibitor (serpin). Source data are provided as a Source data file.

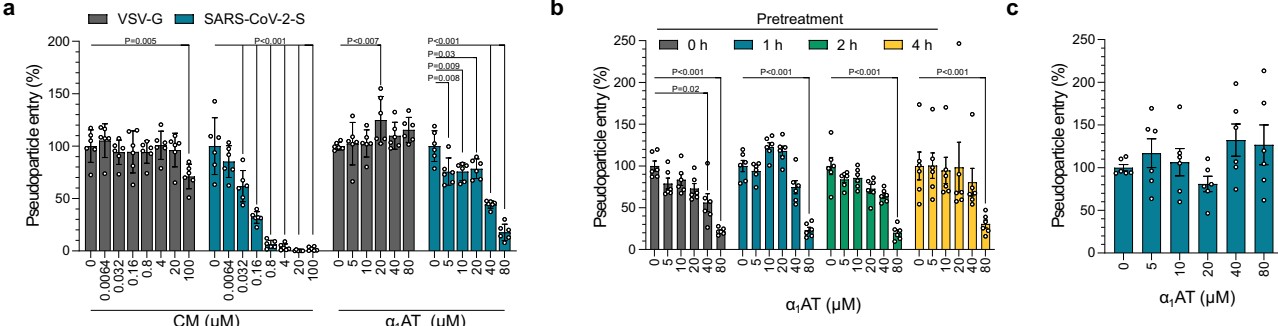

**Fig. 2 $\alpha_1$AT inhibits SARS-CoV-2 spike but not VSV-G mediated pseudovirus entry. a** Prolastin ($\alpha_1$AT) and the control small molecule inhibitor camostat mesylate (CM) were added to Caco2 cells 1 h prior to transduction of cells with rhabdoviral SARS-CoV-2 spike (blue) or VSV-G pseudoparticles (gray). **b** Caco2 cells were treated for the indicated hours with Prolastin ($\alpha_1$AT) and were then transduced with rhabdoviral SARS-CoV-2 spike pseudoparticles. **c** Prolastin ($\alpha_1$AT) was added 2 h post transduction of Caco2 cells with rhabdoviral SARS-CoV-2 spike pseudoparticles. Transduction rates in **a–c** were determined at 16 h after addition of pseudoparticles by measuring luciferase activities in cell lysates. The mean ± SEM from $n = 2$ experiments in biological triplicates are shown (2-way ANOVA with Dunett´s multiple comparison test). Source data are provided as a Source data file.

Analysis of these inhibitory fractions by gel electrophoresis revealed distinct protein bands in sub-fractions 42_55 and 42_57, whereas no protein was detectable in sub-fractions 42_5 to 42_7 (Supplementary Fig. 2a). The most active sub-fraction 42_55 contained a prominent band at ~ 52 kDa, which was also present in other active fractions but hardly found in neighboring fractions showing no antiviral activity (e.g., 42_49) (Supplementary Fig. 2a). This band was excised from the gel, digested with trypsin and subjected to MALDI-TOF-MS revealing a 100 % sequence identity to $\alpha_1$-antitrypsin (SERPINA1) (Supplementary Fig. 2b and Supplementary Table 1), a 52 kDa protease inhibitor[13]. The presence of $\alpha_1$-antitrypsin ($\alpha_1$AT) in inhibitory fractions 42_55 and 42_57 was confirmed by western blot analysis with an $\alpha_1$AT-specific antibody (Supplementary Fig. 2c). $\alpha_1$AT belongs to the serine protease inhibitor (serpin) super-family and protects lung tissue from digestive enzymes released by immune cells, in particular neutrophil elastase[14]. The serpin has a reference range in blood of 0.9–2 mg/ml (corresponding to ~ 17–38 μM), but the concentration can rise 4- to 5-fold upon acute inflammation[15]. $\alpha_1$AT purified from donor blood is also available as pharmaceutical product (e.g., Prolastin) for intravenous substitution therapy of $\alpha_1$AT deficiency, a hereditary disorder that leads to chronic uncontrolled tissue breakdown in the lower respiratory tract[16].

**$\alpha_1$AT inhibits SARS-CoV-2 spike mediated pseudovirus entry.** To test whether the serpin indeed inhibits SARS-CoV-2, Caco2 cells were exposed to Prolastin, a pharmaceutical preparation of $\alpha_1$AT, or camostat mesylate (CM), a small molecule inhibitor of the SARS-CoV-2 spike priming protease TMPRSS2[11,17]. $\alpha_1$AT and CM both suppressed SARS-CoV-2 spike pseudoparticle entry with half-maximal inhibitory concentrations (IC$_{50}$) of ~38.5 μM for $\alpha_1$AT, and ~0.05 μM for CM, respectively (Fig. 2a). Cell viability assays showed that $\alpha_1$AT displayed no cytotoxic effects at concentrations of up to 160 μM (8.3 mg/ml), whereas CM reduced cell viability at concentrations of 200 μM, due to DMSO in the stock (Supplementary Fig. 3). The antiviral activity of $\alpha_1$AT and CM was specific for the coronavirus spike because entry of pseudoparticles carrying the G-protein of VSV was not affected (Fig. 2a). In fact, time-of-addition experiments demonstrated that $\alpha_1$AT prevented single-round SARS-CoV-2 spike pseudovirus entry only if added prior (1–4 h) to or during infection (Fig. 2b) but not if added 2 h post infection (Fig. 2c). Taken together, these data show that $\alpha_1$AT specifically targets SARS-CoV-2 spike-driven entry.

**$\alpha_1$AT inhibits SARS-CoV-2 infection and replication.** To determine whether $\alpha_1$AT inhibits not only spike pseudoparticles but also wild-type SARS-CoV-2, we examined its activity against two SARS-CoV-2 isolates from France (bearing the spike variant D614) and the Netherlands (bearing the fitter spike variant G614)[18,19]. For this, we assessed survival rates of TMPRSS2-expressing Vero E6 cells infected in the absence or presence of EK1, CM or $\alpha_1$AT by MTS assay. In the absence of drugs, infection by both SARS-CoV-2 isolates resulted in virus-induced cytopathic effects (CPE) and reduced cell viability by ~80 % (Fig. 3a and Supplementary Fig. 4a, b). Microscopic evaluation revealed the absence of CPE in the presence of high concentrations of EK1, CM or $\alpha_1$AT, and MTS assay confirmed a concentration-dependent inhibition of cell death and viral replication by EK1 and CM (Supplementary Fig. 4a, b and Fig. 3a) with average IC$_{50}$ values against both SARS-CoV-2 isolates of 2.8 μM for EK1, and 3.6 μM for CM, respectively (Fig. 3a). $\alpha_1$AT inhibited the French SARS-CoV-2 isolate with an IC$_{50}$ of 21.2 μM (1.1 mg/ml), and the Dutch strain with an IC$_{50}$ of 17.3 μM (0.9 mg/ml) (Fig. 3a). Almost complete rescue of cell viability was observed at $\alpha_1$AT concentrations of 40–80 μM (Supplementary Fig. 4a, b and Fig. 3a). A similar antiviral activity of $\alpha_1$AT against both SARS-CoV-2 isolates was determined in Caco2 cells (Supplementary Fig. 4c).

We next performed time-of-addition experiments and found that $\alpha_1$AT most effectively inhibited SARS-CoV-2 replication when added 1 h prior to or simultaneously with infection (Fig. 3b, Supplementary Fig. 5a, b), confirming pseudovirus data (Fig. 2b). $\alpha_1$AT also suppressed SARS-CoV-2 if added 2 or 4 h post infection (Fig. 3b and Supplementary Fig. 5a and b). Considering the 10 h time span of the viral life cycle[20], these data suggest that the serpin inhibits spreading virus infection, i.e., infection of uninfected target cells by progeny virus. Accordingly, no antiviral effect of $\alpha_1$AT was observed when added 24 h post infection, when already two replication cycles have been completed (Fig. 3b and Supplementary Fig. 5a,b). To investigate whether $\alpha_1$AT may inhibit cell-to-cell spread of SARS-CoV-2, a plaque assay with cellulose containing medium (which prevents cell-free viral replication) was performed (Supplementary Fig. 6a,b) and plaque sizes were quantified (Fig. 3c). $\alpha_1$AT concentrations of 50 to 200 μM blocked viral spread most effectively when the serpin was added 1 h prior to infection (54–87 % reduction) or simultaneously with infection (75–89 % reduction). When adding $\alpha_1$AT 1.5 h post infection, spread was still inhibited by 49–83 %, demonstrating effective inhibition of SARS-CoV-2 infection and cell-to-cell viral spread by the serpin.

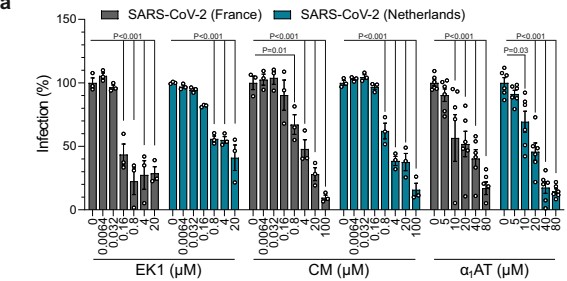

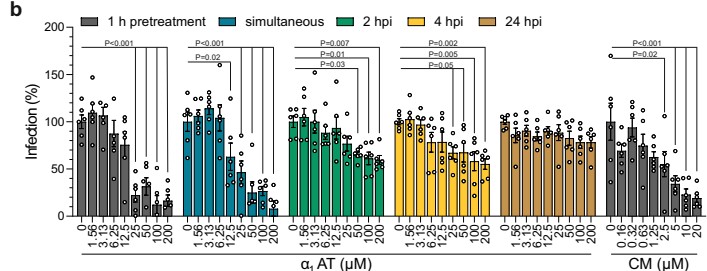

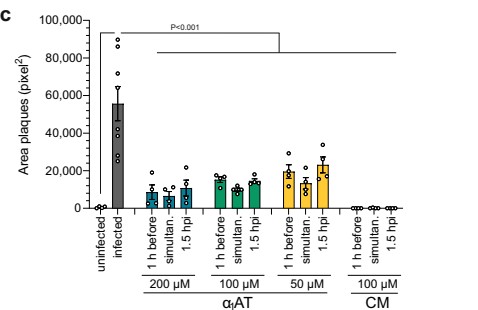

**Fig. 3 α₁AT inhibits SARS-CoV-2 infection and replication. a** TMPRSS2-expressing Vero E6 cells were treated with Prolastin (α₁AT), EK1 or camostat mesylate (CM) for 1 h, and infected with SARS-CoV-2 isolates either from France (gray) or the Netherlands (blue) at a MOI of 0.001. Virus-induced cytopathic effects were assessed at 2 days post infection by MTS assay (see Supplementary Fig. 4 a, b for raw and cell viability data). **b** TMPRSS2-expressing Vero E6 cells were treated with Prolastin (α₁AT) at indicated timepoints prior to, simultaneously with or post infection with SARS-CoV-2 at a MOI of 0.001. Camostat mesylate (CM) control was added 1 h prior to infection. Infection rates were assessed at 2 days post infection by MTS assay (see also Supplementary Fig. 5a, b for raw and cell viability data). **c** TMPRSS2-expressing Vero E6 cells were treated with Prolastin (α₁AT) or CM 1 h prior to, simultaneously with or 1.5 h post infection with SARS-CoV-2. At 1.5 h post infection cellulose overlay was performed. At 2 days post infection, cells were stained with crystal violet (see Supplementary Fig. 6a, b) and plaque areas were quantified. The mean ± SEM from $n = 1$ (**a**, EK1 and CM) or $n = 2$ independent experiments in biological triplicates (**a**, α₁AT, **b**) or duplicates (**c**) and quadruplicates (**c**, infected) are shown. (2-way ANOVA with Dunett´s multiple comparison test (**a**, **b**), ordinary one-way ANOVA with Dunett´s multiple comparison test (**c**)). Source data are provided as a Source data file.

**α₁AT inhibits SARS-CoV-2 replication in primary human airway cells.** To corroborate the antiviral activity of α₁AT in human primary target cells, we used small airway epithelial cells (SAECs) that support low level SARS-CoV-2 replication. For this, SAECs pretreated with buffer only (PBS), 80 μM of α₁AT, or 100 μM of CM were infected with a high dose (MOI of 1) of SARS-CoV-2. Furthermore, α₁AT was added 3 and 24 h post infection to PBS pretreated cells. Cells were cultivated in the presence of the inhibitors for 6 days, and then viral genome copies in supernatants were quantified by RT-qPCR. As shown in Fig. 4a, CM and α₁AT present during infection reduced viral titers by ~92 and 83 %, respectively. α₁AT that was added 3 and 24 h post infection also inhibited viral replication, albeit to a lesser extend (67 and 58 % reduction, respectively). We next analyzed antiviral activity of α₁AT in fully differentiated primary human airway epithelial cells (HAECs) grown at the air–liquid interface. HAECs derived from two donors were treated with 10 μM (0.5 mg/ml) of α₁AT and then exposed to SARS-CoV-2. As control, we used 5 μM of remdesivir, which has previously been shown to suppress coronavirus replication in HAECs[21]. At days 1, 2, and 3 post infection cells were fixed and stained with antibodies against SARS-CoV-2 spike[22], and α-tubulin as marker for ciliated cells at the apical surface[23] (Fig. 4b). In infected, PBS-treated HAECs, SARS-CoV-2 spike expression was readily detectable, mostly in neighboring ciliated cells, and increased between day 2 and 3 (Fig. 4b, c and Supplementary Fig. 7), demonstrating productive infection and viral spread in the epithelia. Spike expression levels in α₁AT and remdesivir treated cultures were greatly reduced at days 2 and 3 in both donors (Fig. 4b, c, Supplementary Fig. 7). Thus, α₁AT suppresses SARS-CoV-2 infection of HAECs.

**α₁AT binds and inhibits TMPRSS2 protease activity.** Finally, we set out to explore the mechanism underlying SARS-CoV-2 inhibition by α₁AT. A recent preprint publication suggests that α₁AT may suppress TMPRSS2, the spike priming protease, similar to camostat mesylate[24]. First, we established a computational model of the Michaelis complex of α₁AT and TMPRSS2 using protein–protein docking calculations complemented by structural refinement (Supplementary Methods) and observed a calculated binding free energy of −10.2 ± 2.2 kcal/mol (Fig. 5a, b). We applied this modeling to the complex of α₁AT (Pittsburgh variant, M358R) and trypsin (S195A) and observed a calculated binding free energy of −10 ± 1.6 kcal/mol (Supplementary Fig. 9). This calculated binding free energy is close to the experimentally determined binding free energy of the α₁AT (Pittsburgh variant, M358R) and trypsin (S195A) complex[25], therefore verifying our modeling approach. The computed structure of the Michaelis complex suggests that this initial step of the reaction mechanism is favored by the interaction of Leu353 and Ala355 of α₁AT (numbers according to PDB ID 3cwm29), with a hydrophobic patch of TMPRSS2 formed by Tyr416, Leu419, and Trp461, which is located next to the catalytic triad (His296, Asp345, Ser441) (Fig. 5b). Thus, the anchoring site, defined by the hydrophobic patch, can effectively position and restrain the reactive center loop prior to the cleavage. These structural insights together with the calculated affinity of this complex, support a favorable interaction between TMPRSS2 and α₁AT. To verify a direct interaction of α₁AT with TMPRSS2, surface plasmon resonance analysis was performed. To this end, recombinant TMPRSS2 was immobilized on a metal surface (Supplementary Fig. 8a) and subjected to increasing doses (0–10 μM) of α₁AT (Supplementary Fig. 8b). A $K_d$ of 941 ± 297 nM was measured by performing equilibrium analysis which was reached within 5 min

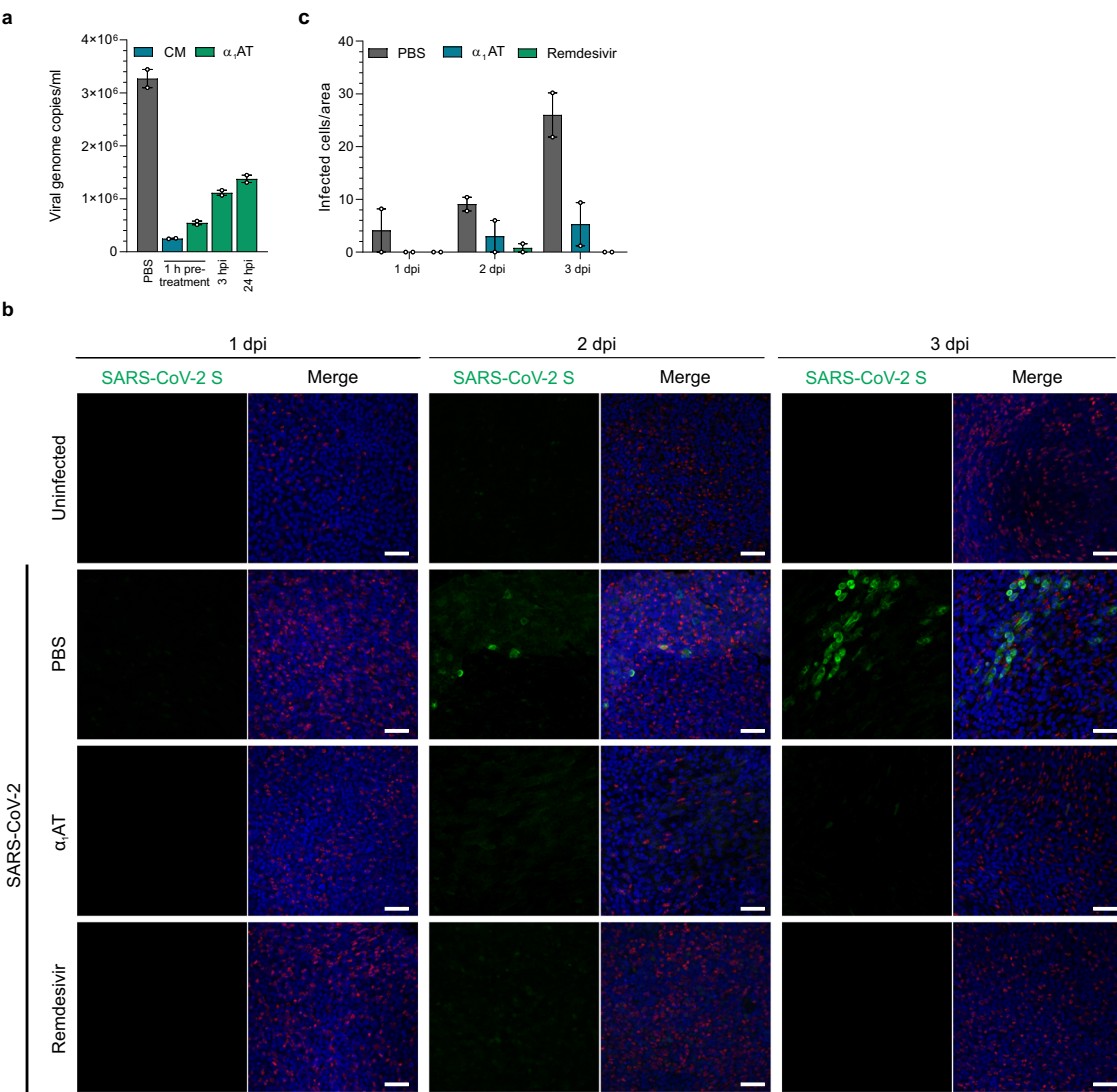

**Fig. 4 $\alpha_1$AT inhibits SARS-CoV-2 replication in primary human airway cells. a** Small airway epithelial cells (SAECs) were treated with 100 µM of Prolastin ($\alpha_1$AT, green) 1 h prior to, or 3 and 24 h post infection (hpi) with SARS-CoV-2 at a MOI of 1. 100 µM CM (blue) added 1 h prior to infection served as control. Immediately after the inoculum was removed (0 dpi) and at day 6 post infection, supernatants were harvested and subjected to RT-qPCR specific for SARS-CoV-2 ORF1b nsp14. Virus titers from day 0 were subtracted from titers at day 6. The means of technical duplicates of one experiment performed in triplicates are shown. **b** The apical and basal site of human airway epithelial cells (HAEC) grown at air–liquid interface was exposed to PBS, $\alpha_1$AT (10 µM or 0.5 mg/ml) and remdesivir (5 µM) and then inoculated with SARS-CoV-2 ($9.25 \times 10^2$ PFU) for 2 h. Cells were fixed at day 1, 2, and 3 post infection, stained with DAPI (cell nuclei, blue), a SARS-CoV-2 specific spike antibody (SARS-CoV-2 S, green) and an $\alpha$-tubulin-specific antibody (red). Images shown are derived from one donor and represent maximum projections of serial sections along the basolateral to apical cell axis. Scale bar: 50 µm. **c** Number of infected cells per area in mock- (PBS, gray), $\alpha_1$AT- (blue) or remdesivir-treated (green), SARS-CoV-2 infected HAECs. Values represent the mean number of infected HAECs from 2 donors at 5 random spots per culture, treatment and day ± SEM. Source data are provided as a Source Data file. For more images, see Supplementary Fig. 7.

interaction time of $\alpha_1$AT with TMPRSS2 (Supplementary Fig. 8c), proving a physical interaction of the protease inhibitor with the protease. To investigate whether $\alpha_1$AT not only binds but also inhibits proteolytic activity of TMPRSS2, the protease was over-expressed in HEK293T cells and incubated with $\alpha_1$AT, CM, or the cysteine protease inhibitor E-64d[11]. Enzymatic activity was assessed by adding a specific substrate that emits fluorescence after proteolytic cleavage. CM but not E-64d suppressed TMPRSS2 activity at sub-micromolar concentrations (Fig. 5c and Supplementary Fig. 8d, f). Interestingly, 100 and 200 µM of $\alpha_1$AT effectively inhibited cell-associated TMPRSS2 activity (Fig. 5c and Supplementary Fig. 8e). Experiments performed with recombinant TMPRSS2 enzyme confirmed effective and dose-dependent inhibition of TMPRSS2 activity by $\alpha_1$AT at physiologically relevant concentrations of 5–50

µM (Fig. 5d). These data demonstrate that the abundant serpin $\alpha_1$AT is an endogenous inhibitor of TMPRSS2 proteolytic activity.

## Discussion

This study demonstrates that $\alpha_1$AT is a potent inhibitor of SARS-CoV-2 which blocks viral replication in cell lines, primary small airway epithelial cells and fully differentiated airway epithelium cultures. $\alpha_1$AT binds and inhibits TMPRSS2, a transmembrane serine protease that cleaves the SARS-CoV-2 spike protein to enable viral fusion. $\alpha_1$AT is the most abundant serine protease inhibitor (serpin) in the circulation (0.9–2 mg/ml, 17–38 µM), and levels of this acute phase protein further increase during acute inflammation[15]. We show that plasma-derived $\alpha_1$AT blocks SARS-CoV-2 infection with $IC_{50}$ values of 10–20 µM, which are

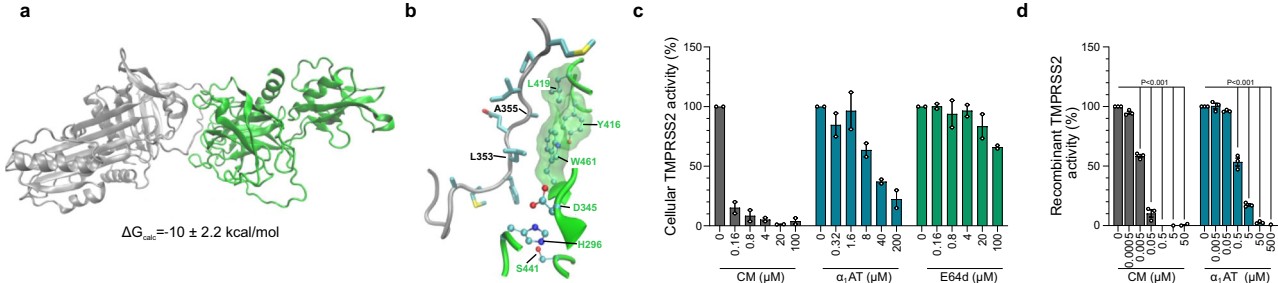

**Fig. 5 α₁AT binds the extracellular region of TMPRSS2 and inhibits TMPRSS2 protease activity. a** Protein–protein docking analysis of a homology model for the TMPRSS2 extracellular fragment (green, PDB 1z8g) and α₁AT (gray, PDB 3cwm) and computationally calculated binding free energy ($\Delta G_{calc}$) of the complex. **b** Detailed view on α₁AT-TMPRSS2 binding interface. The sidechains of α₁AT (gray) residues are represented with sticks, while sidechains of TMPRSS2 (green) are shown with balls and sticks. Hydrogen atoms are omitted for clarity, carbon, oxygen, nitrogen or sulfur atoms of amino acid side chains depicted in light blue, red, dark blue or yellow, respectively. The hydrophobic patch near the TMPRSS2 catalytic triad is highlighted with a green transparent surface. **c** α₁AT inhibits cell-associated TMPRSS2 activity. HEK293T cells were transfected with a TMPRSS2 expression plasmid and treated with Prolastin (α₁AT, blue), camostat mesylate (CM, gray) or E-64d (green) followed by incubation with the fluorogenic TMRPSS2 protease substrate BOC-Gln-Ala-Arg-AMC. Graph shows the relative area under the curve analysis of fluorescence intensities over 2 h that were corrected by values for mock-transfected HEK293T cells. **d** α₁AT inhibits recombinant TMPRSS2 enzyme activity. Recombinant human TMPRSS2 was mixed with Prolastin (α₁AT, blue) or CM (gray) prior to addition of fluorogenic TMPRSS2 protease substrate BOC-Gln-Ala-Arg-AMC, graph shows relative fluorescence intensities after 3 h of incubation. The mean ± SEM of $n = 2$ (**c**) or $n = 3$ (**d**) independent experiments in biological duplicates are shown (ordinary one-way ANOVA with Dunett´s multiple comparison test). Source data are provided as a Source data file.

well within the physiological range, suggesting that plasma itself may exert anti-SARS-CoV-2 activity. In fact, a recent preprint demonstrates that naïve serum exhibits inhibition of SARS-CoV-2 entry and suggested α₁AT and to a lesser degree α-2-macroglobulin as potential antiviral factors[26]. Thus, α₁AT may serve as natural inhibitor of the novel coronavirus, in particular during acute SARS-CoV-2 infection, when blood α₁AT concentrations increase, as recently shown in a cohort of 40 COVID-19 patients[27]. Further studies to clarify the physiological relevance of α₁AT in SARS-CoV-2 infection and whether inter-individual differences in α₁AT concentrations in blood, lungs, and other organs correlate with viral loads and disease progression are thus highly warranted.

α₁AT was isolated as inhibitor of SARS-CoV-2 from a complex peptide/protein library that was generated from pooled bronchoalveolar lavage (BAL). The corresponding fractions presumably contain all soluble peptides and small proteins present in lung fluid and should allow identification of those innate immune factors that are most relevant for controlling viral infection in vivo, at least in the absence of a specific antiviral host immune response. α₁AT-containing fractions 42–45 were more potent in inhibiting SARS-CoV-2 entry than all remaining BAL- and lung-derived fractions, suggesting a relevant role of the serpin in vivo. In normal individuals, α₁AT levels range between 10–40 μM in alveolar interstitial fluid, and 2–5 μM in alveolar extracellular lining fluid[28–30]. Assuming that α₁AT concentrations further increase during the acute phase response, the serpin may act as relevant innate immune factor against SARS-CoV-2 in the respiratory tract.

Biomolecular modeling, surface plasmon resonance, and biochemical analysis established α₁AT as inhibitor of TMPRSS2, a cell surface serine protease that is involved in cell-cell and cell-matrix interactions, and in prostate cancer metastasis[31–33]. TMPRSS2 not only primes the spike protein of SARS-CoV-2 but is also utilized as entry factor by other coronaviruses such as SARS-CoV, MERS-CoV or the common cold corona virus 229E[34–36]. Furthermore, TMPRSS2 is also the major hemagglutinin-activating protease of influenza A virus in human airways[37–39]. It is, therefore, not only of great interest to clarify whether α₁AT may act as broad-spectrum inhibitor against respiratory viral pathogens, but also to evaluate its role in prostate cancer.

α₁AT is an approved drug for treatment of α₁AT deficiency implying repurposing for the therapy of COVID-19. α₁AT deficiency is a hereditary disorder that results in reduced circulating concentrations of α₁AT and consequently a chronic uninhibited breakdown of tissue in the lungs, mainly mediated by neutrophil elastase[40]. Several products containing α₁AT purified from human plasma (such as Prolastin used herein) are approved for decades for intravenous augmentation therapy in patients suffering from α₁AT deficiency. Of note, α₁AT may not only be beneficial in COVID-19 therapy because of its direct antiviral effect by targeting TMPRSS2-mediated SARS-CoV-2 entry, but also by inhibiting neutrophil elastase, which has been proposed to act as alternative spike priming protease[41,42] and might contribute to pulmonary inflammation in COVID-19[43,44]. Moreover, a recent study revealed that the proinflammatory IL-6 to α₁AT ratio in patients with severe COVID-19 was more than two times higher compared to a pneumonia control cohort[27]. Of note, α₁AT can also be administered via inhalation[45] and at substantially higher doses than those in routine α₁AT deficiency. In some studies, α₁AT has been administered at doses of 120 mg/kg[46,47] and even 250 mg/kg without causing side effects, resulting in a 5-fold increase of serpin concentration in lung epithelial lining fluid of α₁AT deficient patients[48]. However, whether α₁AT infusion or inhalation allows to reach local concentrations of the serpin that are sufficient to block SARS-CoV-2 in lungs or other organs without causing severe side effects remains to be addressed in clinical studies. Nevertheless, we and others suggest that α₁AT supplementation may have a therapeutic benefit because of its antiviral and anti-inflammatory properties. Thus, rapid evaluation of α₁AT for the treatment of severe COVID-19 disease is highly warranted[26,27,49,50] and four clinical trials (NCT04495101, NCT04385836, NCT04547140, EudraCT: 2020-001391-15) have been initiated to evaluate the therapeutic potential of α₁AT in hospitalized COVID-19 patients.

## Methods

**Ethic statements.** Ethical approval for the generation of peptide libraries from lungs and BAL was obtained from the Ethics Committee of Ulm University (application numbers 274/12 and 324/12). The collection of tissue and generation of human airway epithelia cell cultures for research from these primary cells has been approved by the ethics committee at the University of Ulm (nasal brushings,

application number 126/19) and Medical School Hannover (airway tissue, application number 2699-2015). Tissues were obtained from donors who gave informed consent.

**Reagents.** Camostat mesylate (SML0057) and E-64d (E8640) were obtained from Merck. α1-antitrypsin (52 kDa) was obtained from Merck (A9024) or GRIFOLS (Prolastin®) and solubilized freshly in dH$_2$O prior to use, Boc-Gln-Ala-Arg-AMC peptide was obtained from PEPTIDE INSTITUTE (3135-v) or Bachem (4017019.0005), EK1 peptide (H$_2$N-SLDQINVTFLDLEYEMKKLEEAIKKLEE-SYIDLKEL-COOH) was synthesized by solid-phase synthesis.

**Generation of a peptide/protein library from lungs.** 6.5 kg of human lung were obtained from deceased individuals without known diseases from pathology of Ulm University. The organs were frozen at −20 °C and freeze-dried. Lung material was homogenized and peptides and small proteins were extracted using ice-cold acetic acid. Then the mixture was centrifuged at 3500 × *g* and the supernatant was applied to a 0.45 μm filter. Thereafter, the obtained peptides and proteins were separated by a 30 kDa molecular weight cut-off ultrafiltration step. The filtrate was separated by reversed-phase (RP) chromatography with a Sepax Poly RP300 50 × 300 mm column (Sepax Technologies, Newark DE, USA 260300-30025) at a flow rate of 100 ml/min and a gradient ranging from 0.1 % TFA (Merck, 1082621000) in ultrapure water (buffer A) to 0.1 % TFA in acetonitrile (J.T.Baker, JT9012-3, buffer B). Reversed-phase chromatographic fractions of 50 ml were collected to constitute the lung peptide bank, from which 1 ml-aliquots (2 %) were lyophilized and used for antiviral testing.

**Generation of a peptide/protein library from BAL.** Clinical samples of bronchoalveolar lavage (BAL) comprising a total of 20 l were collected and immediately frozen for further processing. Peptide/protein extraction was done by acidification with acetic acid to pH 3, followed by centrifugation at 3500 × *g* and filtration (0.45 μm) of the supernatant. Further, the filtered BAL was subjected to a 30 kDa molecular weight cut-off. Chromatographic fractionation of the filtrate was performed by using a reversed-phase (PS/DVB) Sepax Poly RP300 HPLC 30 × 250 mm column (Sepax Technologies, Newark DE, USA 260300-30025), at a flow rate of 55 ml/min and a gradient ranging from 0.1 % TFA in ultrapure water (buffer A) to 0.1 % TFA in acetonitrile. Seventy-three reversed-phase chromatographic fractions of 50 ml were collected to constitute the BAL peptide bank, from which 1 ml aliquots (2 %) were lyophilized and used for antiviral testing. For further purification of active fractions, a second reversed-phase C18 HPLC 4.6 × 250 mm column (Phenomenex 00G-4605-EO) was used at a flow rate of 0.8 ml/min and a gradient ranging from 0.1 % TFA in ultrapure water (buffer A) to 0.1 % TFA in acetonitrile, followed again by lyophilization prior to antiviral testing.

**Cell culture.** Unless stated otherwise, HEK293T cells (ATCC, CRL-3216) were cultivated in DMEM supplemented with 10 % fetal calf serum (FCS), 2 mM L-glutamine, 100 U/ml penicillin, and 100 mg/ml streptomycin. Caco2 cells (ATCC, HTB-37) were cultivated in DMEM supplemented with 10 % FCS, 2 mM glutamine, 100 U/ml penicillin and 100 mg/μl streptomycin, 1× non-essential amino acids (NEAA) and 1 mM sodium pyruvate. TMPRSS2-expressing Vero E6 cells (kindly provided by the National Institute for Biological Standards and Control (NIBSC), #100978) were cultivated in DMEM supplemented with 10 % fetal calf serum (FCS), 2 mM L-glutamine, 100 U/ml penicillin, 100 mg/ml streptomycin and 1 mg/ml geneticin. Human Small Airway Epithelial Cells (Lonza, CC-2547, batch: 18TL082942, donor: 68 years, female) were cultivated in SAGM™ Small Airway Epithelial Cell Growth Medium (Lonza, CC-3118).

**Generation of lentiviral pseudotypes.** For generation of lentiviral SARS-CoV-2 pseudoparticles (LV(Luc)-CoV-2-S) 900,000 HEK293T cells were seeded in 2 ml HEK293T medium. The next day, medium was replaced and cells were transfected with a total of 1 μg DNA using polyethyleneimine (PEI). To this end, 2 % of pCG1-SARS-2-S (encoding the spike protein of SARS-CoV-2 isolate Wuhan-Hu-1, NCBI reference Sequence YP_009724390.1) were mixed with 98 % of pCMVdR8_91 (encoding a replication-deficient lentivirus) and pSEW-Luc2 (encoding a luciferase reporter gene, both kindly provided by Christian Buchholz) in a 1:1 ratio in OptiMEM. Plasmid DNA was mixed with PEI at a DNA:PEI ratio of 1:3 (3 μg PEI per 1 μg DNA), incubated for 20 min at RT and added to cells dropwise. At 8 h post transfection, medium was removed, cells were washed with 2 ml of PBS and 2 ml of HEK293T medium with 2.5 % FCS were added. At 48 h post transfection, pseudoparticles containing supernatants were harvested and clarified by centrifugation for 5 min at 450 × *g*.

**Generation of VSV-based pseudotypes.** For generation of VSV-based SARS-CoV-2 pseudoparticles (VSV(Luc_eGFP)-CoV-2-S), HEK293T were seeded in 30 ml HEK293T medium in a T175 cell culture flask. The next day, cells were transfected with a total 44 μg pCG1-SARS-2-S using PEI. Plasmid DNA and PEI were mixed in 4.5 ml of OptiMEM at a 2:1 ratio (2 μg PEI per 1 μg DNA), incubated for 20 min at RT and added to cells dropwise. 24 h post transfection, medium was replaced and cells were transduced with VSV-G-protein pseudotyped VSV

encoding luciferase and GFP reporter gene (kindly provided by Gert Zimmer, Institute of Virology and Immunology, Mittelhäusern/Switzerland[51]). At 2 h post transduction, cells were washed three times with PBS and cultivated for 16 h in HEPES-buffered HEK293T medium. Virus containing supernatants were then harvested and clarified by centrifugation for 5 min at 450 × *g*, residual pseudo-particles harboring VSV-G-protein were blocked by addition of anti-VSV-G hybridoma supernatant at 1/10 volume ratio (I1, mouse hybridoma supernatant from CRL-2700; ATCC). Virus stocks were concentrated 10-fold using a 100 kDa Amicon molecular weight cutoff and stored at −80 °C until use.

**SARS-CoV-2 strains and propagation.** Viral isolate BetaCoV/France/IDF0372/2020 (#014V-03890) and BetaCoV/Netherlands/01/ NL/2020 (#010V-03903) were obtained from the European Virus Archive global and propagated on Vero E6 or Caco2 cells. To this end, 70–90 % confluent cells in 75 cm$^2$ cell culture flasks were inoculated with SARS-CoV-2 isolate (multiplicity of infection (MOI) of 0.03–0.1) in 3.5 ml serum-free medium. Cells were incubated for 2 h at 37 °C, before adding 20 ml medium containing 15 mM HEPES. Cells were incubated at 37 °C and supernatant harvested when a cytopathic effect (CPE) was visible. Supernatants were centrifuged for 5 min at 1000 × *g* to remove cellular debris, and then aliquoted and stored at −80 °C as virus stocks. Infectious virus titer was determined as plaque-forming units (PFU) on Vero E6 cells, which was used to calculate MOI.

**Screening lung and BAL library for inhibitors of SARS-CoV-2 pseudoparticle entry.** 10,000 Caco2 cells were seeded in 100 μl respective medium in a 96-well flat-bottom plate. The next day, medium was replaced by 40 μl of serum-free medium. For screening peptide containing fractions, 10 μl of the solubilized fraction (in dH$_2$O) were added to cells. Cells were inoculated with 50 μl of infectivity normalized LV(Luc)-CoV2 (or LV(Luc)-no GP control). Transduction rates were assessed by measuring luciferase activity in cell lysates at 48 hours post transduction with a commercially available kit (Promega). Values for untreated controls were set to 100 % transduction.

**Gel electrophoresis and western blotting.** Gel electrophoresis of active fractions was performed on a 4–12 % Bis–Tris protein gel (NuPAGE™). Prior to electrophoresis, samples were reduced with 50 mM β-mercaptoethanol and heated for 10 min at 90 °C. The gel was either directly stained with Coomassie G-250 (GelCode™ Blue Stain) or blotted on PVDF membranes. Blotted membranes were stained with a polyclonal anti-α$_1$AT antibody (1:1000; Proteintech 16382-1-AP). After incubation with IRDye anti-rabbit secondary antibodies (1:20,000; LiCor), the staining was visualized using an Odyssey Infrared Imager (Licor).

**Tryptic in-gel digestion of proteins.** Bands of interest were excised and the proteins were digested with trypsin. Tryptic peptides were eluted from the gel slices with 1 % trifluoric acid.

**Matrix-assisted laser-desorption ionization time-of-flight mass spectrometry (MALDI-TOF-MS).** MALDI-TOF-MS was performed on an Ultraflex TOF/TOF mass spectrometer (Bruker Daltonics, Bremen) equipped with a nitrogen laser and a LIFT-MS/MS facility. The instrument was operated in the positive-ion reflectron mode using 2.5-dihydroxybenzoic acid and methylendiphosphonic acid as matrix. Sum spectra consisting of 200–400 single spectra were acquired. For data processing and instrument control the Compass 1.4 software package consisting of FlexControl 4.4, FlexAnalysis 3.4 4, Sequence Editor and BioTools 3.2 and ProteinScape 3.1. were used. External calibration was performed with a peptide standard (Bruker Daltonics).

**Database search.** Proteins were identified by MASCOT peptide mass fingerprint search (http://www.matrixscience.com) using the Uniprot Human database (version 20200226, 210438 sequence entries; *p* < 0.05). For the search, a mass tolerance of 75 ppm was allowed and oxidation of methionine as variable modification was used.

**Pseudoparticle inhibition experiments.** One day prior to transduction, 10,000 Caco2 cells were seeded in 100 μl respective medium in a 96-well plate. For the addition of α$_1$AT prior to infection, medium was replaced by 80 μl of serum-free medium and cells were incubated with serial dilutions of Prolastin (α$_1$AT) for 0, 1, 2, and 4 h at 37 °C followed by infection with 20 μl of infectivity normalized VSV (Luc)-CoV-2-S pseudoparticles. To investigate whether α$_1$AT acts post viral entry, cells were inoculated with 20 μl of infectivity normalized VSV(Luc)-CoV-2-S pseudoparticles. After 2 h, cells were washed with 100 μl PBS and 100 μl serum-free medium as well as 20 μl of serially diluted α$_1$AT were added. Transduction rates were assessed by measuring luciferase activity in cell lysates at 16 h post transduction with a commercially available kit (Promega). Values for untreated controls were set to 100 % transduction.

**TMPRSS2 activity measurement.** 20,000 HEK 293T cells were seeded in 100 μl of the respective medium in a 96-well flat-bottom plate. The next day, cells were

transfected with 100 ng of TMPRSS2 (addgene 53887, kindly provided by Roger Reeves, Johns Hopkins University, Baltimore, United States) per well using PEI transfection reagent. Briefly, plasmid DNA was mixed with PEI in a 3:1 ratio in serum-free medium, incubated for 20 min at RT and added to cells dropwise. At 12 h post transfection, medium was removed and 60 μl of PBS were added, followed by serial dilutions of Prolastin or inhibitor control. After incubation for 15 min at 37 °C, 20 μl of protease substrate BOC-Gln-Ala-Arg-AMC were added. Fluorescence intensity was recorded at an excitation wavelength of 380 nm and emission wavelength of 460 nm in 1 min intervals for 2 h at 37 °C in a Synergy™ H1 microplate reader (BioTek, USA) with Gen 5 3.04 software. For assessing the activity of recombinant human TMPRSS2, 25 μl of serially diluted Prolastin or inhibitor control in assay buffer (50 mM Tris-HCL, 0.154 mM NaCl pH 8.0) were incubated with 25 μl of recombinant TMPRSS2 enzyme for 10 min at 37 °C, followed by addition of 20 μM BOC-Gln-Ala-Arg-AMC protease substrate. Fluorescence intensity was measured after 3 h at an excitation wavelength of 360 nm and emission wavelength of 465 nm in a Tecan Genios with Magellan V6.4 software.

**Binding free energy calculations**. The Central Limit Free Energy Perturbation (CL-FEP) approach[52] was employed to estimate the binding affinities of the complexes of the enzymes TMPRSS2 and trypsin with $\alpha_1$AT. The structure of the trypsin-$\alpha_1$AT complex reported with PDB ID 1OPH[53] was used for the calculations, while the complex TMPRSS2-$\alpha_1$AT was modeled via docking and subjected to further structural refinement, see computational details in supplementary materials. CL-FEP allows an unbiased end-state calculation of free energy changes directly from explicit solvent simulations. The sampling was performed with molecular dynamics simulations of the individual proteins, the enzyme-$\alpha_1$AT complexes and the bulk solvent. The proteins were sampled under wall-type restraints on their bound-state conformations, which permits to focus the sampling on the most relevant states to the binding energy. The force constant for these restraints was 100 kcalmol$^{-1}$ Å$^{-2}$ and a restraint-free range of 5 Å was allowed with respect to the initial conformation. 100 ns of sampling simulations were collected for each simulation box, with the energy sampled every 5 ps. The simulation boxes and molecular dynamics setup were obtained using the CLFEP-GUI web server (https://clfep.zmb.uni-due.de/). The CL-FEP analyses were performed with ten checkpoints containing increasing fractions of the total energy samples, an oversampling ratio of osr = 20 was fixed to bring the free energy variance until the level of (kT)[2], and the second-order cumulant estimator (C2) was evaluated to determine the free energy changes at each checkpoint. The final estimate corresponds to the average among the converged checkpoints. The error is obtained as the standard deviation among the individual estimations. The convergence analyses[52] for all the checkpoints are summarized in Supplementary Tables 2 and 3.

**Surface plasmon resonance**. SPR assay was carried out on a Reichert SR7000 system using SPRAutolink 1.1.9 software. To measure the binding affinity between TMPRSS2 and $\alpha_1$AT, TMPRSS2 (Creative BioMart) was immobilized on a 11-mercaptoundecanoic acid (MUA) functionalized gold sensor chip through covalent coupling of amine groups. The surface was activated using 0.4 M N-hydroxysuccinimide (NHS) and 0.1 M 1-ethyl-3-(3-dimethylaminopropyl)carbodiimide (EDC) coupling reagents and the TMPRSS2 protein was coupled at a concentration of 12.5 μg/ml in 10 mM acetate buffer at pH = 5.5. Residual activated carboxylic groups were blocked using a 1 M ethanolamine solution at pH = 8.5. $\alpha_1$AT was dissolved at different concentrations (0, 0.2, 0.5, 1, 2, 5, 10, and 20 μM) in HEPES buffer (10 mM HEPES, 150 mM NaCl, 0.05 % Tween-20; pH = 7.5) and introduced with a flow rate of 10 μl/min. The data were analyzed with Prism 7 using non-linear regression fit and the value for the equilibrium dissociation constant $K_d$ was obtained by fitting the response at equilibrium against the concentration. Successful immobilization of TMPRSS2 was verified using a TMPRSS2 antibody (rabbit anti-human IgG, ThermoFischer Scientific, PA5-14264) dissolved in HEPES buffer to a concentration of 0.2 mg/ml (1:10 of 2 mg/ml stock). As a negative control, TMPRSS2 antibody solution was passed through a channel with no TMPRSS2 protein immobilization.

**Cytotoxicity assay**. To assess cytotoxicity of $\alpha_1$AT and camostat mesylate (CM), 10,000 Caco2 cells were seeded in 100 μl medium in a 96-well flat-bottom plate. The next day, medium was replaced by 80 μl of serum-free Caco2 medium and cells were treated with serial dilutions of Prolastin, CM or DMSO as solvent control for CM. After 48 h, cell viability was assessed by measuring ATP levels in cells lysates with a commercially available kit (CellTiter-Glo®, Promega).

**Virus-induced cytopathic effect analysis**. To quantify SARS-CoV-2 wildtype infection, virus-induced cell death was inferred from remaining cell viability determined by MTS (3-(4,5-dimethylthiazol-2-yl)-5-(3-carboxymethoxyphenyl)-2-(4-sulfophenyl)-2H-tetrazolium) assay. To this end, 20,000 TMPRSS2-expressing Vero E6 or 30,000 Caco2 cells were seeded in 96-well plates in 100 μl respective medium. The next day, medium was replaced with serum-free medium and the respective compound of interest was added. After incubation for 1 h at 37 °C the cells were infected with a MOI of 0.001 of the SARS-CoV-2BetaCoV/France/IDF0372/2020 or BetaCoV/Netherlands/01/NL/2020, in a total volume of 180 μl.

2 (TMPRSS2-expresing Vero E6) or 3 (Caco2) days post infection, infection was quantified by detecting remaining metabolic activity. To this end, 36 μl of CellTiter 96® AQueous One Solution Reagent (Promega G3580) were added to the medium and incubated for 3 h at 37 °C. Then, optical density (OD) was recorded at 620 nm using an Asys Expert 96 UV microplate reader (Biochrom)with DigiRead 1.26 software. To determine infection rates, sample values were subtracted from untreated control and untreated control set to 100 %. For time of addition experiments, 18,000 TMPRSS2-expressing Vero E6 cells were seeded in 100 μl respective medium. The next day, respective compounds were added 1 h prior to, simultaneously with, or 2, 4 and 24 h post infection with SARS-CoV-2 (BetaCoV/France/IDF0372/2020) at a MOI of 0.01. Viral inoculum was removed 2 h post infection and cells were washed twice with PBS. Cell viability was assessed at 2 days post infection by MTS assay as described above.

**Plaque assay**. 700,000 TMPRSS2-expressing Vero E6 cells were seeded in 12-well plates in 1 ml of the respective medium. The next day, medium was refreshed and cells were treated with respective compound 1 h prior to, simultaneously with or 1.5 h post infection with SARS-CoV-2 (BetaCoV/Netherlands/01/NL/2020). After infection, cells were overlaid with respective medium supplemented with cellulose. At 2 days post infection, cells were fixed in 4 % paraformaldehyde in PBS for 45 min, washed one with PBS and stained with 0.5 % crystal violet in 0.1 % triton in H$_2$O for 30 min. Staining solution was removed and cells were washed three times with water. Virus-induced plaques were quantified in ImageJ 1.53c by assessing number of pixels occupied by plaques.

**Infection of small airway epithelial cells**. 30,000 SAEC were seeded in 100 μl of respective medium in 96-well flat bottom plate. The next day, medium was refreshed and Prolastin was added 1 h prior to or 3 h and 24 h post infection with SARS-CoV-2 (BetaCoV/France/IDF0372/2020) at a MOI of 1 for 3 h. Cells were washed three times with PBS and fresh medium was added. Immediately after the wash step (0 dpi) and at 6 dpi, supernatant samples were collected for qPCR analysis. To analyze virus replication by RT-qPCR, RNA was isolated using the QIAamp Viral RNA Mini Kit (Qiagen) according to the manufacturer's instructions. For RT-qPCR, samples were thawed and 5 μl of lysate used as sample in a 20 μl reaction using Fast Virus 1-Step Mastermix (Thermo Fisher, # 4444436), 0.5 μM Taqman primers targeting SARS-CoV-2-ORF1b-nsp14 and 0.25 μM probe (for primer sequences see Supplementary Table 4). Cycling conditions in were as follows: 1 cycle of reverse transcription (50 °C, 300 s) and RT-inactivation (95 °C, 20 s); 40 cycles of denaturation (95 °C, 5 s) and extension (60 °C, 30 s) in a Step One Plus qPCR cycler (AppliedBiosystems) with Step One Software 2.3. RNA isolated from virus stocks of BetaCoV/France/IDF0372/2020 with copy numbers previously estimated using synthetic SARS-CoV-2 RNA standard (Twist Bioscience, #102024) was used to determine genome copies from Ct values.

**Generation of human airway epithelial cells**. Differentiated air–liquid interface cultures of human airway epithelial cells (HAECs) were generated from primary human basal cells isolated from airway epithelia. Cells were expanded in a T75 flask (Sarstedt) in Airway Epithelial Cell Basal Medium supplemented with Airway Epithelial Cell Growth Medium SupplementPack (both Promocell). Growth medium was replaced every two days. Upon reaching 90 % confluence, HAECs were detached using DetachKIT (Promocell) and seeded into 6.5 mm Transwell filters (Corning Costar). Filters were precoated with Collagen Solution (StemCell Technologies) overnight and irradiated with UV light for 30 min before cell seeding for collagen crosslinking and sterilization. $3.5 \times 10^4$ cells in 200 μl growth medium were added to the apical side of each filter, and an additional 600 μl of growth medium was added basolaterally. The apical medium was replaced after 48 h. After 72–96 h, when cells reached confluence, the apical medium was removed and basolateral medium was switched to differentiation medium. Differentiation medium consisted of a 1:1 mixture of DMEM-H and LHC Basal (Thermo Fisher) supplemented with Airway Epithelial Cell Growth Medium SupplementPack and was replaced every 2 days. Air-lifting (removal of apical medium) defined day 0 of air–liquid interface (ALI) culture, and cells were grown at ALI conditions until experiments were performed at day 25–28. To avoid mucus accumulation on the apical side, HAEC cultures were washed apically with PBS for 30 min every 3 days from day 14 onwards.

**SARS-CoV-2 infection of HAECs**. Immediately before infection, the apical surface of HAECs grown on Transwell filters were washed three times with 200 μl PBS to remove accumulated mucus. Then, 10 μM of $\alpha_1$AT or 5 μM remdesivir were added into the basal medium and onto the apical surface. Cells were infected with $9.25 \times 10^2$ plaque-forming units (PFU) of SARS-CoV-2 (BetaCoV/France/IDF0372/2020). After incubation for 2 h at 37 °C, viral inoculum was removed and cells were washed three times with 200 μl PBS and again cultured at the air–liquid interface. At 1, 2, and 3 days post infection, cells were fixed for 30 min in 4 % paraformaldehyde in PBS, permeabilized for 10 min with 0.2 % saponin and 10 % FCS in PBS, washed twice with PBS and stained with anti-SARS-CoV-2 spike (ab252690, Abcam) and anti-alpha-tubulin (MA1-8007, Thermo Scientific) diluted 1:300 to 1:500, respectively, in PBS, 0.2 % saponin and 10 % FCS over night at 4 °C.

Subsequently, cells were washed twice with PBS and incubated for 1 h at room temperature in PBS, 0.2 % saponin and 10 % FCS containing AlexaFluor 488-labeled anti-rabbit and AlexaFluor 647-labeled anti-rat secondary antibody, respectively (all 1:500; Thermo Scientific) and DAPI + phalloidin AF 405 (1:5000; Thermo Scientific). Images were taken on an inverted confocal microscope (Leica TCS SP5, Leica Microsystems, Leica application suite version 2.7.3.9723) using a ×40 lens (Leica HC PL APO CS2 40 × 1.25 OIL). Images for the blue (DAPI), green (AlexaFluor 488) and far-red (AlexaFluor 647) channels were taken in sequential mode using appropriate excitation and emission settings that were kept constant for all the acquisitions. For quantification, randomly chosen locations in each filter were selected and z-stacks were acquired. A maximum z projection was performed and anti-SARS-CoV-2 positive cells per area ($0.15 \, mm^2$) were visually identified and counted.

**Non-linear regression and statistics**. Unless stated otherwise, analysis was performed using GraphPad Prism version 8.4.2. Calculation of $IC_{50}$ values via non-linear regression was performed using normalized response-variable slope equation. For statistical analysis, either ordinary one-way ANOVA or 2-way ANOVA with Dunett´s multiple comparison test was used as indicated in respective figure legends.

**Reporting summary**. Further information on research design is available in the Nature Research Reporting Summary linked to this article.

## Data availability

Proteins were identified by MASCOT peptide mass fingerprint search (http://www.matrixscience.com) using the Uniprot Human database (version 20200226, 210438 sequence entries; $p < 0.05$), Human Uniprot database, P01009 (A1AT_HUMAN). Crystal structures were obtained from SWISSMODEL repository (https://swissmodel.expasy.org/repository/uniprot/O15393?csm=C05B5531C8A311C7) or Protein Data Bank with accession codes PDB ID: 1OPH, PDB ID: 1Z8G, PDB ID: 3CWM. Source data are provided with this paper.

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

## Acknowledgements

L.W., C.C., T.W., R.G., M.H., and A.S. are part of and R.G. is funded by a scholarship from the International Graduate School in Molecular Medicine Ulm. This work was supported by the German Research Foundation (DFG) through "Fokus-Förderung COVID-19" grants to J.M. (MU 3115-13), A.K. (KL 2544/8-1) and M.F. (FR 2974/3-1), the CRC 1279 to S.S., F.K., E.S-G., and J.M., SPP1923 to F.K. and K.S., and SP1600/4-1 to K.S. J.M. also acknowledges funding by the EU's Horizon 2020 research and innovation programme (Fight-nCoV, 101003555 to J.M.). J.A.M. is indebted to the Baden-Württemberg Stiftung for the financial support by the Eliteprogramme for Postdocs and receives funding by the DFG. S.P. was supported by BMBF (RAPID Consortium, 01KI1723D and 01KI2006D; RENACO, 01KI20328A, 01KI20396). E.S-G. further acknowledges the Boehringer Ingelheim Foundation (Plus-3 Grant) and the Deutsche Forschungsgemeinschaft (DFG, German Research Foundation) under Germany's Federal and State Excellence Strategy EXC- 2033 Projektnummer 390677874 and the Collaborative Research Center CRC 1093. J.A.M., J.M., F.K., M.F. and A.K. further acknowledge funding by the Sonderfördermassnahme COVID-19 of the MWK Baden-Württemberg, Germany, and A.K. by the Heisenberg-Programm (KL 2544/6-1). K.S. was supported by the BMBF (IMMUNOMOD, 01KI2014).

## Author contributions

L.W. generated LV pseudotypes, performed screening, gel electrophoresis, α₁AT inhibition studies with pseudotypes and in-cell TMPRSS2 activity measurements; C.C., T.W., and J.A.M generated SARS-CoV-2 stocks und performed all infection experiments in the BSL-3; F.Z., C.P.B., R.G., and A.S. generated VSV pseudotypes; T.W. supported L.W. in most experiments; M.H. performed western blots; C.C. and S.K. supported L.W. in screening; N.P. synthesized EK1, generated peptide libraries and purification; G.F. and R.L. generated HAEC and did staining; P.K. and V.M. performed SPR and provided reagents and controls; Y.B.R.B. and E.S.G. designed, performed and analyzed the results of the computational modeling; S.P. and M.H. performed recombinant TMPRSS2 activity analysis; L.S. supervised the generation of peptide libraries; D.R.T. performed autopsies and collected lungs; C.S. collected BAL; S.S. supervised BSL-3 work; L.W. arranged the figures and L.W., C.C., T.W, J.A.M., and R.G. edited the manuscript; A.K. advised and wrote the manuscript; G.L. performed mass spectrometry; K.S. advised and edited the manuscript; F.K. supervised work and wrote the manuscript; B.M. supervised statistics; M.F. supervised work with HAEC; J.M. is responsible for the study, supervised all work and wrote the manuscript.

## Funding

## Competing interests

The authors declare no competing interests.
